

# Using electroretinograms and multi-model inference to identify spectral classes of photoreceptors and relative opsin expression levels

Nicolas Lessios[1,2]

[1] School of Life Sciences, Arizona State University, Tempe, AZ, USA
[2] Current affiliation: Department of Neuroscience, University of Arizona, Tucson, AZ, USA

## ABSTRACT

Understanding how individual photoreceptor cells factor in the spectral sensitivity of a visual system is essential to explain how they contribute to the visual ecology of the animal in question. Existing methods that model the absorption of visual pigments use templates which correspond closely to data from thin cross-sections of photoreceptor cells. However, few modeling approaches use a single framework to incorporate physical parameters of real photoreceptors, which can be fused, and can form vertical tiers. Akaike's information criterion ($AIC_c$) was used here to select absorptance models of multiple classes of photoreceptor cells that maximize information, given visual system spectral sensitivity data obtained using extracellular electroretinograms and structural parameters obtained by histological methods. This framework was first used to select among alternative hypotheses of photoreceptor number. It identified spectral classes from a range of dark-adapted visual systems which have between one and four spectral photoreceptor classes. These were the velvet worm, *Principapillatus hitoyensis*, the branchiopod water flea, *Daphnia magna*, normal humans, and humans with enhanced S-cone syndrome, a condition in which S-cone frequency is increased due to mutations in a transcription factor that controls photoreceptor expression. Data from the Asian swallowtail, *Papilio xuthus*, which has at least five main spectral photoreceptor classes in its compound eyes, were included to illustrate potential effects of model over-simplification on multi-model inference. The multi-model framework was then used with parameters of spectral photoreceptor classes and the structural photoreceptor array kept constant. The goal was to map relative opsin expression to visual pigment concentration. It identified relative opsin expression differences for two populations of the bluefin killifish, *Lucania goodei*. The modeling approach presented here will be useful in selecting the most likely alternative hypotheses of opsin-based spectral photoreceptor classes, using relative opsin expression and extracellular electroretinography.

Corresponding author
Nicolas Lessios,
nlessios@email.arizona.edu

## INTRODUCTION

Animals possess a diversity of opsin proteins, one of the main genetic components underlying spectral photoreceptor classes (*Porter et al., 2012*). It is now possible to identify functional amino acid sequence sites of opsin proteins that determine the spectral sensitivity of photoreceptors (*Arendt et al., 2004*; *Porter et al., 2007*). The number and wavelength sensitivity of spectral photoreceptor classes an organism possesses is needed to understand whether it can discriminate natural spectra (i.e., has some form of color vision), and also to understand the mechanistic context of visually guided behavior (*Kelber & Osorio, 2010*). Spectral classes of photoreceptors are generally identified using a combination of extracellular and intracellular electroretinographic (ERG) techniques (*Arikawa, Inokuma & Eguchi, 1987*). Extracellular recordings detect a summed contribution of multiple classes of photoreceptors, including relatively rare classes that are difficult to identify using intracellular techniques. It is possible to isolate spectral photoreceptor classes using chromatic adaptation, where light of a restricted waveband is used to light-adapt single photoreceptor classes and the resulting effects on spectral sensitivity are observed in extracellular recordings. However, because visual pigments are all natively sensitive to short wavelengths (*Bowmaker, 1999*), this procedure is most applicable to long wavelength receptors in organisms that possess up to three spectral photoreceptor classes (*Goldsmith, 1986*). Intracellular techniques are the most accurate for verifying the existence of spectral classes; but they can be further supported by modeling approaches which incorporate physical parameters obtained from histological techniques (*Stavenga & Arikawa, 2011*).

I have developed a framework of multi-model selection using overall spectral sensitivities of the visual system. The goals of this framework were to:

A) Identify the most likely number of opsin-based spectral photoreceptor classes of visual systems from extracellular ERGs, and from known parameters of the photoreceptor array.
B) Establish whether differences between individuals in structural photoreceptor parameters affect identification of the same underlying number of opsin-based spectral photoreceptor classes found in A.
C) Map relative opsin expression levels to relative visual pigment concentrations when structural parameters and opsin identities of the photoreceptor array are known.

The framework used here employs Akaike's information criterion ($AIC_c$) to select among competing alternative hypotheses (*Akaike, 1974*). AIC is an objective measure that imposes a realistic penalty for over-parameterization (*Burnham & Anderson, 2002*). For goals A) and B) the alternative hypotheses are the number and relative area in cross-section, or frequency, of spectral photoreceptor classes. For goal C), the alternative hypotheses are the number of opsins which differ in relative expression level. Others have used multi-model selection to identify the number of photoreceptors in the eyes of oceanic fish, using the relative contributions of photoreceptor classes in cross-section to absorbance (*Horodysky et al., 2008*, *2010*). Existing models of absorptance, which use
parameters of real photoreceptors (*Snyder, Menzel & Laughlin, 1973*), are developed here to incorporate parameters of multiple tiers, or to model absorptive layers affecting the spectral sensitivity of underlying photoreceptors.

## MATERIALS AND METHODS

### Visual modeling of photoreceptor absorptance

The fused photoreceptor array per unit length was modeled as

$$\xi_j(\lambda) = \sum \alpha_i(\lambda) \frac{A_i}{A} k, \tag{1}$$

where $\alpha_i$ is the normalized absorption spectrum of each rhodopsin visual pigment, $A_i/A$ is the relative area or frequency in cross-section of each photoreceptor $i$, and $k$ is the peak absorption coefficient. Values used for $k$ for invertebrates (0.008 $\mu m^{-1}$) were established by *Bruno, Barnes & Goldsmith (1977)* and are typical for crustaceans and insects (*Cronin et al., 2014*). Values used for $k$ for humans (0.015 $\mu m^{-1}$) are typical for vertebrates (*Wyszecki & Stiles, 1982*). Absorptance of a tiered photoreceptor array, composed of $j$ tiers was calculated as follows:

$$S(\lambda) = \sum \left( T_{(j-1)} \left( 1 - e^{-\xi_j(\lambda) l_j} \right) \right) \tag{2}$$

where $T_{j-1}$ is the transmittance through all preceding vertical tiers ($T_0 = 1.0$ for the first tier). Normalized absorbance templates developed by *Stavenga, Smits & Hoenders (1993)*, referred to here as SSH, and by *Govardovskii et al. (2000)*, referred to here as GFKRD, were used for visual pigment absorption spectra $\alpha_i$, each of which has a wavelength of peak absorbance $\lambda_{max}$. Normalized absorption templates have two primary components, an alpha band with a wavelength of peak absorbance that is determined by the interaction between the chromophore and the opsin protein, and a beta band which absorbs in the UV, and is mainly determined by the chromophore itself (*Bowmaker, 1999*). Effects of including both alpha and beta bands were assessed in a preliminary analysis of a global model, then only alpha bands were considered (see $AIC_c$ procedure). $S(\lambda)$ was normalized to 1 as in *Stavenga & Arikawa (2011)*.

### Example selection

I used organisms which have between one and five classes of spectral photoreceptors to examine capabilities and limitations of the described framework. Four organisms were used to address goals A) and B), and spectral sensitivities from dark-adapted eyes were used to minimize effects of variation among individuals of changing visual pigment concentration, pigment migration, or varying levels of metarhodopsin (*Stavenga, 2010*). The fifth organism was used to address goal C) to map differences in visual pigment concentrations to relative opsin expression level for two populations of the same species.

1. The onycophoran velvet worm, *Principapillatus hitoyensis* (Fig. 1A) expresses a single spectral opsin class in its photoreceptors (*Beckmann et al., 2015*).
2. *Homo sapiens* possesses one rod and three cone (S, M, and L) photoreceptor classes. Normal human scotopic sensitivity (Fig. 1B) is represented by S-class cone and rod

photoreceptor sensitivities (*Bowmaker & Dartnall, 1980*; *Wyszecki & Stiles, 2000*). In contrast, scotopic sensitivity of patients with enhanced S-cone syndrome (Fig. 1C) is a condition in which S-cone frequency is increased due to mutations in a transcription factor that controls photoreceptor expression (*Haider et al., 2000*). Human absorptance models are corrected here for transmittance through the lens and a distal macula tier protecting the retina that affects spectral sensitivity (*Wyszecki & Stiles, 1982*).

3. The branchiopod crustacean water flea, *Daphnia magna* (Fig. 1D) possesses four spectral photoreceptor classes (*Smith & Macagno, 1990*).

4. The swallowtail butterfly, *Papilio xuthus* (Figs. 1E and 1F) possesses at least five main spectral classes of photoreceptor type (*Arikawa, Inokuma & Eguchi, 1987*), in several classes of ommatidia with specialized filtering pigments (*Stavenga & Arikawa, 2011*).

5. The bluefin killifish, *Lucania goodei*, possesses five cone photoreceptor classes based on known opsins (SWS1, SWS2B, SWS2A, RH2-1, and LWS). Separate populations of this species have been shown to regulate opsin expression depending on their photic environments (*Fuller et al., 2004*). Killifish absorptance models are corrected here for transmittance through a tier of distal ellipsosomes associated with cone classes found in the related killifish *Fundulus heteroclitus* (*Flamarique & Harosi, 2000*), and through the lens of the Nile tilapia *Oreochromis niloticus* (*Lisney, Studd & Hawryshyn, 2010*). The relative frequency of the cones cone classes that express SWS2B, RH2-1, and LWS were corrected to take into account that they are double cones.

## Data extraction, binning, and averaging from multiple recording locations

Published spectral sensitivity data were extracted using GetData v.2.26 (*Fedorov, 2013*) from *Arikawa, Inokuma & Eguchi (1987)*, *Smith & Macagno (1990)*, *Jacobson et al. (1990)*, *Fuller et al. (2003)* and *Beckmann et al. (2015)*. Where needed, units were converted from log sensitivity to relative sensitivity. Preliminary analysis indicated that 20 and 10 nm wavelength intervals provided identical results. Binning was therefore carried out at 20 nm intervals for all sensitivity data. Sensitivity ranges were 410–690 nm for humans, 350–690 nm for *Principapillatus hitoyensis* and *D. magna*, and 310–690 nm for *P. xuthus*. For *P. xuthus* (*Arikawa, Inokuma & Eguchi, 1987*) had recorded extracellularly from multiple regions of the compound eye (dorsal, medial, and ventral). Binned sensitivities from each region were therefore averaged to provide a single relative spectral sensitivity (Figs. 1E and 1F).

## Incorporating known photoreceptor lengths $l_j$ in Eq. (2)

Photoreceptor lengths were estimated or taken from published sources: *Principapillatus hitoyensis* (100 μm) (*Beckmann et al., 2015*); *H. sapiens* parafovea (22.5 μm) (*Bowmaker & Dartnall, 1980*; *Cronin et al., 2014*); *D. magna* (12.0 μm) (*Smith & Macagno, 1990*); *Papilio xuthus* (500 μm) (*Arikawa & Stavenga, 1997*); and *L. goodei* (18 μm) (*Moldstad, 2008*). The fused cross-sectional and tiered three-dimensional photoreceptor array is known for *D. magna* and for *P. xuthus*: as in many insects and crustaceans

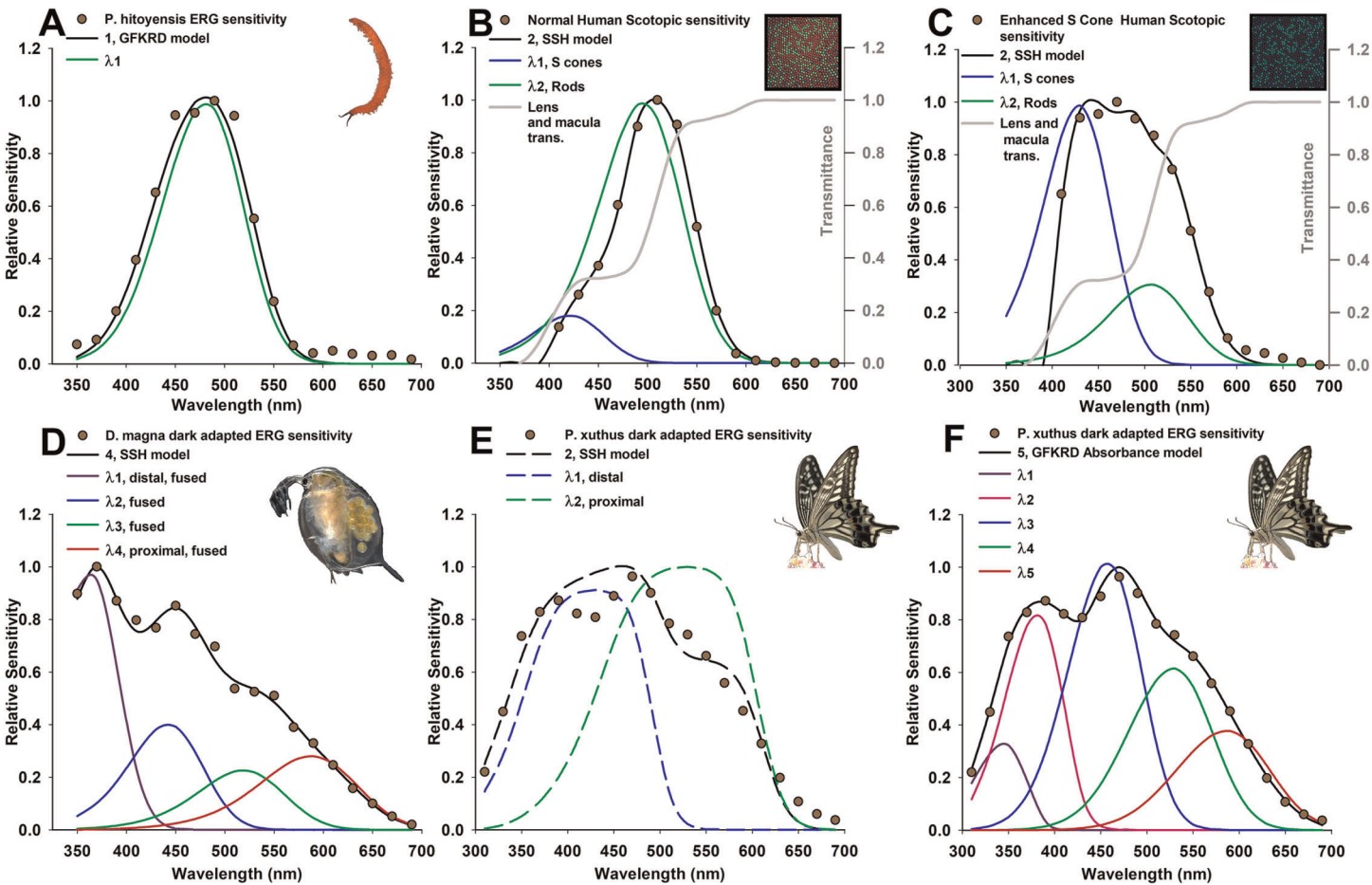

**Figure 1 Photoreceptor absorptance models (curves) based on known photoreceptor lengths and vertical tiering, fit to relative spectral sensitivity data extracted from published sources (data points).** Models were selected using Akaike's information criterion corrected for small sample sizes (AIC$_c$) with the best three models shown in Tables 1 and 2, and all models in Tables S1 and S2. (A) Velvet worm *Principapillatus hitoyensis* sensitivity, known to be represented by a single spectral opsin class expressed in its photoreceptors (*Beckmann et al., 2015*). (B and C) Normal and enhanced S-cone human scotopic sensitivities, known for normal humans to be represented by S-class cone and rod photoreceptor sensitivities, and with a higher frequency of S cones in patients that have enhanced S-cone syndrome (*Jacobson et al., 1990*; *Hood et al., 1995*; *Haider et al., 2000*). Absorptance models for humans are corrected for transmittance through the lens and a distal macula layer which protects the retina, but which does not contribute to spectral sensitivity (gray lines) (*Wyszecki & Stiles, 2000*). (D) *Daphnia magna* sensitivity, known to be represented by four spectral photoreceptor classes with a distal UV receptor (*Smith & Macagno, 1990*). (E and F) *Papilio xuthus* sensitivity, averaged from extracellular recordings from multiple positions in the compound eye, known to be represented by at least five main spectral photoreceptor classes (*Arikawa, Inokuma & Eguchi, 1987*). (E) Absorptance models (dashed lines) illustrate poor results with this technique because of model over-simplification explained in text. (F) Absorbance (given by Eq. (1)) at a cross-section approximately two-thirds from the distal tip of the rhabdom of an ommatidium selects five spectral photoreceptor classes, with deviations of each spectral class explained further in the text due to specialized filtering pigments.

(*Kelber & Henze, 2013*), the shortest wavelength receptor of both species becomes axon-like partway through the optical unit. Models considered here for *D. magna* and *P. xuthus* which have more than one spectral class of photoreceptor incorporate this structure in Eq. (2), and in the optimization procedure. The shortest wavelength receptor of *D. magna* ommatidia forms a fused structure in the distal (upper) half of the optical unit (6.0 µm), with a short-wavelength receptor replaced by a longer-wavelength sensitive receptor in the proximal (lower) half of the optical unit (6.0 µm).
The distal two-thirds of the optical unit (333 μm) of *P. xuthus* ommatidia is modeled as a single optical unit, replaced by a long wavelength receptor in the proximal portion (167 μm).

## Parameter estimates, maximum likelihood estimation, optimization, and AIC$_c$ procedure

The maximum likelihood estimate (MLE) of each model was calculated according to *Burnham & Anderson (2002)*

$$log(L(\hat{\underline{\Theta}})) = -\frac{1}{2}log(\hat{\underline{\sigma}^2}) - \frac{n}{2}log(2\pi) - \frac{n}{2}, \tag{3}$$

where the MLE for $\hat{\sigma}^2$ is $\frac{RSS}{n}$, and *RSS* is the residual sum of squares for a given model. Optimization of model parameters $\lambda_{max}$, and $A_i/A$ for goals A) and B), then *k* for goal C) were carried out using custom scripts, and the optimization toolbox in MATLAB. A linear constraint was used for *D. magna* and *P. xuthus* during optimization to maintain $\lambda_{max1}$ as the shortest wavelength receptor in the first tier ($\lambda_{max\ i} < \lambda_{max\ i+1}$). The absorption coefficients for *L. goodei* were constrained to a value greater than 0.001/μm and less than 1.000/μm.

I used AIC$_c$ for small samples to compare the optimized log-likelihood,

$$\text{AIC}_c = -2\log\left(L(\hat{\underline{\Theta}}) + \frac{2K(K+1)}{n-K-1}\right) \tag{4}$$

where *K* is the number of parameters.

AIC scores were compared to the best model ($\Delta\text{AIC}_c = \text{AIC} - \min\text{AIC}$), and were weighted using Akaike weights

$$w\text{AIC}_c = e^{-0.5\Delta\text{AIC}_i} \bigg/ \left(\sum_1^R e^{-0.5\Delta\text{AIC}_r}\right), \tag{5}$$

where *R* is the number of models considered. $w\text{AIC}_c$ provides a weighting indicating the likelihood of a single optimized model compared to all considered models, while penalizing for over-parameterization. Akaike weights were used to calculate evidence ratios relative to the best model (Tables 1 and 2; Tables S1 and S2). See *Posada & Buckley (2004)* and *Symonds & Moussalli (2011)* for abbreviated explanations of Akaike weights and evidence ratios.

The above procedure was first used to optimize models to extracellular ERG data for *D. magna*. Beta bands were considered for every possible photoreceptor, an "all subsets" generalized linear model examining the influence of each parameter on $S(\lambda)$ relative to known $S(\lambda)$, comparing among 124 optimized models (Table S4). Generalized linear model results indicated beta bands were uninformative for model selection as they were the least important covariate $\beta$, in this case $(\frac{\hat{\beta}_\beta}{E(yi)}) < 3.0$, and upon removal led to a reduction in AIC$_c$ according to methods outlined in *Burnham & Anderson (2002)* and *Arnold (2010)*. Models which included beta bands were therefore removed and only models in Tables S1–S3 were included for the formal analysis.

**Table 1 Absorptance model comparisons for *Principapillatus hitoyensis* and *Homo sapiens* using maximum likelihood and Akaike's information criterion corrected for small sample sizes (AIC_c).**

| Species or condition | Reference<br><br>Model | $\lambda_{max1}$ $(A_1/A)$ | $\lambda_{max2}$ $(A_2/A)$ | $\lambda_{max3}$ $(A_3/A)$ | $\lambda_{max4}$ $(A_4/A)$ | AIC_c | $\Delta$AIC_c | wAIC_c | Evidence ratio |
|---|---|---|---|---|---|---|---|---|---|
| *Principapillatus hitoyensis* | *Beckmann et al. (2015)* | 484 | – | – | – | – | – | – | – |
| | 1, GFKRD | 481 (1.0) | – | – | – | 55.8 | 0 | 0.508 | – |
| | 1, SSH[a] | 481 (1.0) | – | – | – | 54.9 | 0.863 | 0.330 | 1.54 |
| | 2, GFKRD[b] | 481 (0.70) | 481 (0.30) | – | – | 53.2 | 2.54 | 0.143 | 3.56 |
| Normal human (scotopic) | *Wyszecki & Stiles (2000)* | 420 | 497 | – | – | – | – | – | – |
| | 2, SSH | 421 (0.16) | 495 (0.85) | – | – | 91.3 | 0 | 0.500 | – |
| | 2, GFKRD[a] | 419 (0.17) | 495 (0.83) | – | – | 91.1 | 0.176 | 0.458 | 1.09 |
| | 3, SSH[b] | 407 (0.11) | 493 (0.45) | 493 (0.45) | – | 85.1 | 6.24 | 0.02 | 22.6 |
| Enhanced S-cone human (scotopic) | *Jacobson et al. (1990)* | 420 | 497 | – | – | – | – | – | – |
| | 2, SSH | 429 (0.76) | 506 (0.24) | – | – | 65.6 | 0 | 0.587 | – |
| | 2, GFKRD[a] | 429 (0.75) | 506 (0.25) | – | – | 64.0 | 1.62 | 0.261 | 2.25 |
| | 3, GFKRD[b] | 375 (0.27) | 432 (0.54) | 507 (0.20) | – | 62.0 | 3.79 | 0.088 | 6.65 |

**Notes:**
Photoreceptor arrays were modeled for each species and condition using parameters from Eqs. (1) and (2) (Materials and Methods). $A_i/A$, relative area of photoreceptor in cross-section. SSH, rhodopsin visual pigment template (*Stavenga, Smits & Hoenders, 1993*). GFKRD, rhodopsin visual pigment template (*Govardovskii et al., 2000*). Three best-supported models are displayed here for each species or condition. All model comparisons considered are included in Table S1. Evidence ratios were calculated relative to the best model for each species or condition.
[a] Models with ambiguous wAIC_c (evidence ratio < 2.0).
[b] Models with low support relative to the best model (evidence ratio > 2.0).

## RESULTS AND DISCUSSION

Visual physiologists have long used inferences from thin sections to identify the wavelength of peak absorbance for visual pigments. The reason is the absorbance of visual pigments can be predicted very accurately once the wavelength of peak absorbance, $\lambda_{max}$, is identified. In practice, this is achieved by excising a portion of the retina, taking sections of the photoreceptors, and measuring the fraction of light which is transmitted or absorbed. Ideally, this is performed on single photoreceptors, using a range of narrow-bandwidth light to infer the wavelength of peak absorbance. Vision researchers found that peak absorbance can be used to normalize the rest of the absorbance curve to create a template curve (*Dartnall, 1953*). Then, using just the wavelength of peak absorbance, it was found that the rest of the curve can be predicted using mathematical expressions. These nomograms correspond closely to visual pigment that is extracted in solution (*Govardovskii et al., 2000*). Therefore, the idea of a "universal visual pigment template" is very useful when the wavelength of peak absorbance is known, referred to as "normalized absorption templates". And because $\lambda_{max}$ of a visual pigment is primarily determined by the particular opsin amino acids in opsin–chromophore interactions, it is now possible to specify which amino acids determine a specific absorbance profile (*Arendt et al., 2004*; *Porter et al., 2007*). However, a normalized absorption template can be misleading when placing the function of a single photoreceptor class in context of other photoreceptors, or the overall spectral sensitivity of the eye. Therefore, absorptance models were used here with the assumption that they are a more realistic approximation

**Table 2 Absorptance model comparisons for *Daphnia magna* and *Papilio xuthus* using maximum likelihood and Akaike's information criterion corrected for small sample sizes (AIC$_c$).**

| Species or condition | Reference / Model | $\lambda_{max1}$ ($A_1/A$) | $\lambda_{max2}$ ($A_2/A$) | $\lambda_{max3}$ ($A_3/A$) | $\lambda_{max4}$ ($A_4/A$) | $\lambda_{max5}$ ($A_5/A$) | AIC$_c$ | $\Delta$AIC$_c$ | $w$AIC$_c$ | Evidence ratio |
|---|---|---|---|---|---|---|---|---|---|---|
| *Daphnia magna* (tiered absorptance) | *Smith & Macagno (1990)* | 356 | 440 | 521 | 592 | – | – | – | – | – |
| | 4, SSH | 362 (0.52) | 442 (0.21) | 518 (0.12) | 587 (0.15) | – | 46.2 | 0 | 0.979 | – |
| | 3, SSH[b] | 367 (0.50) | 455 (0.22) | 560 (0.28) | – | – | 38.3 | 7.96 | 0.018 | 53.64 |
| | 4, GFKRD[b] | 364 (0.50) | 437 (0.21) | 508 (0.12) | 582 (0.17) | – | 33.3 | 12.97 | <0.01 | 656 |
| *Papilio xuthus* (tiered absorptance) | *Arikawa, Inokuma & Eguchi (1987)* | 360 | 390/400 | 460 | 520 | 600 | – | – | – | – |
| | 2, SSH | 429 (0.48) | 529 (0.52) | – | – | – | 34.9 | 0 | 0.726 | – |
| | 3, SSH[b] | 429 (0.56) | 505 (0.23) | 559 (0.21) | – | – | 31.4 | 3.477 | 0.128 | 5.69 |
| | 2, GFKRD[b] | 422 (0.49) | 529 (0.51) | – | – | – | 30.5 | 4.389 | 0.081 | 8.98 |
| *Papilio xuthus* (absorbance) | *Arikawa, Inokuma & Eguchi (1987)* | 360 | 390/400 | 460 | 520 | 600 | – | – | – | – |
| | 5, GFKRD | 346 (0.10) | 381 (0.25) | 457 (0.32) | 529 (0.20) | 586 (0.12) | 50.4 | 0 | 0.653 | – |
| | 3, SSH[b] | 371 (0.35) | 463 (0.37) | 557 (0.28) | – | – | 47.8 | 2.63 | 0.176 | 3.71 |
| | 4, GFKRD[b] | 348 (0.13) | 385 (0.26) | 465 (0.36) | 559 (0.25) | – | 46.6 | 3.83 | 0.096 | 6.77 |

Notes:
Tiered photoreceptor arrays were modeled for each species and condition using parameters from Eqs. (1) and (2) (Materials and Methods). $A_i/A$, relative area of photoreceptor in cross-section. SSH, rhodopsin visual pigment template (*Stavenga, Smits & Hoenders, 1993*). GFKRD, rhodopsin visual pigment template (*Govardovskii et al., 2000*). Three best-supported models are displayed here for each species or condition. All model comparisons considered are included in Table S2. Evidence ratios were calculated relative to the best model for each species or condition.
[a] Models with ambiguous $w$AIC$_c$ (evidence ratio < 2.0).
[b] Models with low support relative to the best model (evidence ratio > 2.0).

for overall sensitivity estimated from extracellular ERGs, and to incorporate multiple layers of filtering.

The first goal of the framework presented here was to find whether overall sensitivity can be used to identify the most likely number of underlying spectral classes of photoreceptors. As can be seen from the fit of each best model to the data (Fig. 1), and from the evidence ratios (Tables 1 and 2), the framework described here is generally able to resolve the number and relative cross-sectional area or frequency of the photoreceptors in the visual systems I have modeled. It is important to note that AIC avoids over-parameterization with the clearest example shown here for velvet worm *Principapillatus hitoyensis*. Though one to five spectral classes were considered (Table 1; Table S1), to add parameters (i.e., more complex models), the likelihood of those models, given the data, must outweigh the penalty imposed by additional parameters. *Principapillatus hitoyensis* sensitivity (Fig. 1A, points) is represented by a single spectral opsin class expressed in its photoreceptors with an estimated $\lambda_{max}$ of 484 nm, and the best-supported model here was a single receptor GFKRD absorptance model with $\lambda_{max}$ of 481 nm (Fig. 1A, black curve).

This framework is also able to resolve the presence of more photoreceptors, if the data support them. *Daphnia magna* sensitivity (Fig. 1D) is represented by four spectral photoreceptor classes with a distal UV receptor (*Smith & Macagno, 1990*), and the best-supported model here was a four receptor SSH absorptance model (Table 2; Table S2). The results strongly support the presence of a UV sensitive photoreceptor in

**Table 3  AIC inferences compared to traditional hypothesis testing which uses an F-test to distinguish between two best models of similar fit.**

| Species or condition | Model | Residual sum of squares (RSS) | F-test comparing two models with best fit | p Value from F-test | Number of parameters (K) | Evidence ratio |
|---|---|---|---|---|---|---|
| *Principapillatus hitoyensis* | 1, GFKRD | 0.031 | 1.90 | 0.13 | 3 | – |
| | 2, GFKRD | 0.024 | – | – | 5 | 3.56 |
| Normal human (scotopic) | 2, SSH | 0.003 | 2.75 | 0.05* | 5 | – |
| | 3, SSH | 0.002 | – | – | 7 | 22.6 |
| Enhanced S-cone human (scotopic) | 2, SSH | 0.012 | 2.75 | 0.05* | 5 | – |
| | 3, GFKRD | 0.008 | – | – | 7 | 6.65 |
| *D. magna* | 4, SSH | 0.009 | 11 | <0.001 | 9 | – |
| | 3, SSH | 0.031 | – | – | 7 | 53.64 |
| *Papilio xuthus* (tiered absorptance) | 2, SSH | 0.100 | 2.05 | 0.10 | 5 | – |
| | 3, SSH | 0.076 | – | – | 7 | 5.69 |
| *Papilio xuthus* (absorbance) | 5, GFKRD | 0.006 | 10.5 | <0.001 | 11 | – |
| | 3, SSH | 0.034 | – | – | 7 | 3.71 |

**Notes:**
The best model and the closest model with a different number of photoreceptor spectral classes according to AIC are displayed in this order for each species or condition. An F-test typically used for comparing non-linear regression models with similar fits was used here to compare two models with lowest residual sum of squares. In cases were $p < 0.05$, the model with more parameters is accepted. Examples which deviated from AIC results are shown with an asterisk (*). This comparison indicates that AIC provides a similar framework to non-linear regression to compare multiple models and can generally eliminate unneeded parameters (in this table, photoreceptor classes and cross-sectional area).

the compound eye of *D. magna*. Though it was poorly supported in comparison to the best model (evidence ratio > 2.0), the second best-supported model for *D. magna* is a three receptor SSH model, rather than a four receptor GFKRD model (Table 2). This finding can be explained by better performance of the SSH template in the UV range, which has been documented (*Stavenga, 2010*). Future modeling efforts for organisms with UV photoreceptors should expect stronger cumulative performance of absorptance models based on the SSH template.

Results for *Principapillatus hitoyensis* and *D. magna* indicate that this technique resolves a range of opsin-based photoreceptor classes in visual systems. In comparison to more traditional null-hypothesis testing (Table 3), AIC results were similar, with the exception of humans, in which an F-test of non-linear regression results would identify three spectral photoreceptor classes. Table 3 also shows how the penalty imposed by AIC for unneeded parameters provides similar results to comparisons of non-linear regression models. Intuitively, this type of multi-model selection should make sense in terms of natural selection, as maintaining photoreceptors is costly, and if they do not match natural spectra, there is an inarguable cost. It should also be emphasized that, to date, *P. hitoyensis* and *D. magna* have not been found to possess specialized optical filtering in their visual systems (*Smith & Macagno, 1990*; *Martin, 1992*; *Beckmann et al., 2015*).

To establish whether this framework can identify the same number and photoreceptor $\lambda_{max}$ of a visual system when the frequency of the spectral photoreceptor classes is known to differ between individuals, this framework was applied to scotopic human spectral sensitivities. Normal and enhanced S-cone human scotopic sensitivities (Figs. 1B and 1C) are represented by S-cone and rod photoreceptors, with a higher frequency of S cones in patients with enhanced S-cone syndrome (*Jacobson et al., 1990*;

**Table 4 Photoreceptor parameters and reported relative opsin expression values for two populations of *L. goodei* used in modeling absorption coefficient *k* for known opsin-based spectral photoreceptor classes.**

| Species and population | $\lambda_{max1}$ $(A_1/A)$ | Opsin$_1$ (exp) | $\lambda_{max2}$ $(A_2/A)$ | Opsin$_2$ (exp) | $\lambda_{max3}$ $(A_3/A)$ | Opsin$_3$ (exp) | $\lambda_{max4}$ $(A_4/A)$ | Opsin$_4$ (exp) | $\lambda_{max5}$ $(A_5/A)$ | Opsin$_5$ (exp) |
|---|---|---|---|---|---|---|---|---|---|---|
| *L. goodei* spring population | 359 (0.08) | SWS1 (0.21) | 405 (0.31) | SWS2B (0.26) | 454 (0.16) | SWS2A (<0.01) | 538 (0.25) | RH2-1 (0.27) | 572 (0.25) | LWS (0.25) |
| *L. goodei* swamp population | 359 (<0.01) | SWS1 (0.11) | 405 (0.16) | SWS2B (0.21) | 456 (0.10) | SWS2A (<0.01) | 541 (0.32) | RH2-1 (0.33) | 573 (0.42) | LWS (0.34) |

**Notes:**
Values for $\lambda_{max}$ and cone frequencies ($A_i/A$) were identified using microspectrophotometry (*Fuller et al., 2003*). These values were incorporated as constants into model optimization of absorption coefficients below. Relative opsin expression (exp) is in comparison to the sum of all opsins expression is reported from *Fuller et al. (2004)*. Relative expression levels should be compared to Table 5 normalized absorption coefficients.

*Hood et al., 1995; Haider et al., 2000*). Although the full width half-maximum (FWHM) of normal, dark-adapted humans is 20 nm narrower than *Principapillatus hitoyensis* (Fig. 1), the best-supported model using this technique is a two receptor GFKRD absorptance model (Table 1). The narrow bandwidth of normal dark-adapted humans can be explained primarily by the presence of the macula, and illustrates that overlooking absorptive layers which affect spectral sensitivity of underlying photoreceptors leads to erroneous interpretation of the number of spectral photoreceptor classes they possess. As can be seen from Table 1 and Fig. 1, the framework presented here identifies increased frequency of S cones in individuals with enhanced S-cone syndrome, and also identifies two primary spectral photoreceptor classes.

To identify limitations of model over-simplification, I applied this technique to *P. xuthus* sensitivity (Figs. 1E and 1F). Absorptance models (Fig. 1E, dashed lines) illustrate poor results with this technique for *P. xuthus*: as can be seen by the very broad (>100 µm at FWHM) sensitivity of each modeled photoreceptor in the "best" model, self-screening has been over-estimated. *P. xuthus* is known to use specialized filtering pigments in part to sharpen the spectral sensitivity of its receptors (*Arikawa, 2003*). Opsins are expressed heterogeneously in separate classes of ommatidia leading to regions of their compound eyes differing in spectral sensitivity (*Arikawa, Inokuma & Eguchi, 1987*; *Arikawa & Stavenga, 1997*). However, absorbance (Fig. 1F) at cross-section two-thirds from the distal tip of the rhabdom of an ommatidium selects a five spectral photoreceptor GFKRD absorbance model. *P. xuthus* possesses filtering pigments in the peak spectral regions of the photoreceptor classes with the largest deviations identified by this technique ($\lambda_{max1}$, $\lambda_{max2}$, and $\lambda_{max5}$, Table 2). *P. xuthus* is not known to possess filtering pigments in the peak bandwidths of the remaining spectral classes ($\lambda_{max3}$ and $\lambda_{max4}$, Table 2) (*Wakakuwa, Stavenga & Arikawa, 2007*). The comparison of *P. xuthus* absorbance and absorptance results serves to illustrate that multi-model selection must be used judiciously based on what is known for a given visual system. Absorbance results presented here fail to identify the diversity of receptors, and ommatidial spectral classes of organisms where fine-scale spectral discrimination is essential to their visual ecology (*Koshitaka et al., 2008*). The modeling framework is still useful for incorporating both electrophysiology and histology to compare the effects on overall spectral sensitivity. Deviations from these models can identify the presence of previously unknown spectral

**Table 5 Absorptance model comparisons for two populations of *L. goodei* identify differences in absorption coefficient *k* for known opsin-based spectral photoreceptor classes.**

| Species and population | Model | SWS1 $k_1$ $(k_1/k)$ | SWS2B $k_2$ $(k_2/k)$ | SWS2A $k_3$ $(k_3/k)$ | RH2-1 $k_4$ $(k_4/k)$ | LWS $k_5$ $(k_5/k)$ | AIC$_c$ | ΔAIC$_c$ | wAIC$_c$ | Evidence ratio |
|---|---|---|---|---|---|---|---|---|---|---|
| *L. goodei* spring population | 3, SSH | – (–) | 0.0045 (0.40) | – (–) | 0.0042 (0.37) | 0.0027 (0.24) | 37.8 | 0 | 0.448 | – |
| | 3, GFKRD[a] | – (–) | 0.019 (0.42) | – (–) | 0.017 (0.38) | 0.0095 (0.21) | 37.0 | 0.819 | 0.298 | 1.51 |
| | 4, SSH[a] | 0.0030 (0.18) | 0.0051 (0.32) | – (–) | 0.0050 (0.31) | 0.0032 (0.20) | 36.7 | 1.18 | 0.249 | 1.80 |
| *L. goodei* swamp population | 3, SSH | – (–) | 0.0027 (0.28) | – (–) | 0.0036 (0.38) | 0.0033 (0.34) | 37.0 | 0 | 0.945 | – |
| | 3, GFKRD[b] | – (–) | 0.0077 (0.33) | – (–) | 0.0085 (0.36) | 0.0074 (0.31) | 30.2 | 6.833 | 0.031 | 30.46 |
| | 2, SSH[b] | – (–) | – (–) | – (–) | 0.011 (0.54) | 0.0092 (0.46) | 28.6 | 8.42 | 0.014 | 67.38 |

**Notes:**
Three best-supported models are reported for comparison between absorption coefficients ($k$) normalized by the sum of absorption coefficients ($k_i/k$). All model comparisons considered are included in Table S3. Evidence ratios were calculated relative to the best model for each species or condition.
[a] Models with ambiguous wAIC$_c$ (evidence ratio < 2.0).
[b] Models with low support relative to the best model (evidence ratio > 2.0).

filters for an organism, or can provide objective multi-model inference to validate what is known of their visual system.

The examples used until this point are from dark-adapted eyes, and $k$, the peak absorption coefficient in Eq. (2), remained constant. In these examples $\lambda_{max}$, the wavelength of peak absorbance of each photoreceptor, and $A_i/A$, the relative area or frequency in cross-section of each photoreceptor, were allowed to vary for optimization. However, relative opsin gene expression levels can vary over short time scales (*Fuller & Claricoates, 2011*), or can change depending on light environment (*Fuller, Noa & Strellner, 2010*). Therefore, an additional goal of the modeling framework presented here was to use overall sensitivity to map relative opsin expression levels to visual pigment concentration in an organism with well-characterized photoreceptor classes, by allowing $k$ to vary. The bluefin killifish, *L. goodei*, was used as two populations found in spring (broad wavelength) and swamp (red-shifted) light environments have been shown to differ in relative opsin expression level for multiple cone photoreceptor classes. The first two rows of Table 4 show the known values of $\lambda_{max}$, and $A_i/A$ which were entered as constants into this framework, and the final two rows show the expression level of each opsin in proportion to all other opsins which were measured in a real-time PCR study (*Fuller et al., 2004*).

The alternative hypotheses in this example pertained to the number of photoreceptors that had visual pigments with absorption coefficients $k < 0.001/\mu m$. The three best models for the spring population are all well supported by the data (evidence ratio < 2.0), indicating that the framework presented here will select the presence of photoreceptors with three or four visual pigments in meaningful concentrations; the model with three visual pigments is supported for the swamp population (Table 5). Though killifish are

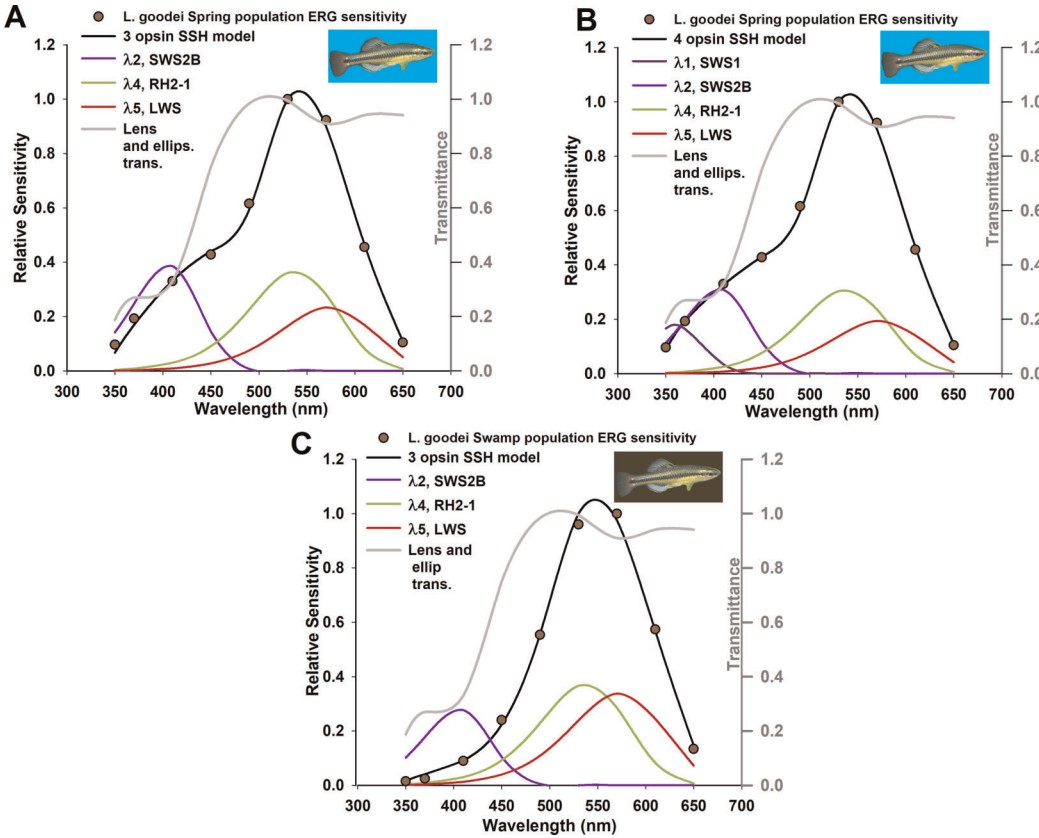

**Figure 2 Absorption coefficient models based on known relative opsin expression levels from two populations for the killifish, _Lucania goodei_.** Models were fit to relative spectral sensitivity data extracted from published sources (data points). Models were selected using Akaike's information criterion corrected for small sample sizes ($AIC_c$) with the best three models shown in Tables 1 and 2, and all models in Table S3. $\lambda_{max}$ and $A_i/A$ were held constant and not included as parameters.

known to have at least five main spectral cone photoreceptor classes, relative expression levels of class SWS2A reported to date for this species are not found at meaningful expression levels (Table 4) (_Fuller et al., 2004_). The relative frequency of UV photoreceptors (which express opsin SWS) for swamp populations is less than 0.01 (Table 4), indicating that three visual pigments are likely the main contributors to overall sensitivity. The best SSH models and transmittance through the lens and ellipsosomes are shown in Fig. 2. The optimized values of $k$ for each visual pigment were also informative. Though they tended to individually be less than values typically found in vertebrate photoreceptors, the sum of these ranges from 0.0163 in the best four SSH model, to ~0.0455 in one of three GFKRD models. These are all within the range of $k$ typically found in vertebrate photoreceptors (_Cronin et al., 2014_). These values are informative for two reasons: first, they mean that there are most likely physiological limits to visual pigment concentrations because they are near saturation in photoreceptors, and second, when modeling $k$ it is assumed to be at the peak wavelength of each visual pigment, which is not possible at all wavelengths, which has been addressed by _Warrant & Nilsson (1998)_. Further, when $k$ is compared to the sum

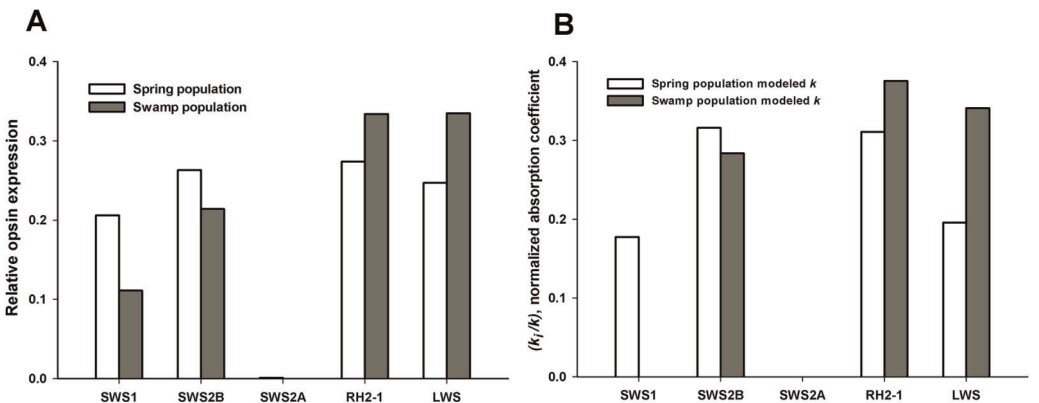

**Figure 3 Absorption coefficient values from** Table 5 **for comparison to relative opsin expression levels from** *Fuller et al. (2004)*. Opsin expression was quantified relative to the total opsin expression level.

of all $k$ values in Fig. 3, it becomes apparent that the main opsin expression results have been reproduced by these optimized models. This indicates that future opsin expression studies, which are often difficult to place in context of either overall sensitivity or behavior (*Fuller & Noa, 2010*) could use the framework suggested here, and models of overall sensitivity inferred from extracellular ERGS.

Currently, empirical studies which identify the spectral properties of individual photoreceptor cells or visual pigments are difficult to place in the larger context of the visual system if all the organism's spectral classes are not identified. The framework I have presented here can be informative for future opsin expression studies and for objectively guiding extracellular or intracellular electroretinography.

## ACKNOWLEDGEMENTS

I thank Justin Marshall and one anonymous reviewer for their reviews of a previous version of this manuscript, as well as the editor, Magnus Johnson. Thanks to Ronald Rutowski and Jonathan Cohen for support throughout the course of this research.

### Funding

This work was supported by the National Science Foundation Graduate Research Fellowship under Grant No. DGE-0802261 and an NIH IRACDA PERT fellowship through the Center for Insect Science (K12 GM000708) at the University of Arizona. The funders had no role in study design, data collection and analysis, decision to publish, or preparation of the manuscript.

### Grant Disclosures

The following grant information was disclosed by the authors:
National Science Foundation Graduate Research Fellowship: No. DGE-0802261.
NIH IRACDA PERT fellowship through the Center for Insect Science: K12 GM000708.
## Competing Interests

The authors declare that they have no competing interests.

## Author Contributions

- Nicolas Lessios conceived and designed the experiments, performed the experiments, analyzed the data, contributed reagents/materials/analysis tools, wrote the paper, prepared figures and/or tables, and reviewed drafts of the paper.

## Data Availability

The raw data has been supplied as Supplemental Dataset Files.

## Supplemental Information

Supplemental information for this article can be found online at http://dx.doi.org/10.7717/peerj.3595#supplemental-information.

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
