# Peer review of "Using electroretinograms and multi-model inference to identify spectral classes of photoreceptors and relative opsin expression levels"

_PeerJ, doi:10.7717/peerj.3595_

## Round 0.1 · original submission · Major Revisions

Both reviewers indicate that they feel this is a paper worthy of publication but would like to see a bit more clarity and more background added to give the paper broader appeal.

Reviewer 1 ·

Basic reporting

This paper presents a method to discriminate between different photoreceptor arrangements in animal vision. The paper is extremely technical for both workers in the research field of ecology and those active in vision. The paper would gain much in accessibility if considerable background was added, at least on visual photoreceptors and the appropriate terminology of absorbance, absorptance, alpha and beta bands.

Experimental design

OK

Validity of the findings

OK

Additional comments

This paper presents a method to discriminate between different photoreceptor arrangements in animal vision. The paper is extremely technical for both workers in the research field of ecology and those active in vision. The paper would gain much in accessibility if considerable background was added, at least on visual photoreceptors and the appropriate terminology of absorbance, absorptance, alpha and beta bands.

Abstract
L3 Change ‘absorbance’ into ‘absorption’ (check definition of absorbance)
L9 Whether or not this technique ‘correctly selects’ will be a matter of discussion: delete. That whole sentence is weird, or, I don’t understand how classes can be identified for different visual systems.
L65 ‘this procedure is applicable for long wavelength receptors in organisms that possess three spectral photoreceptor classes or less’ – The procedure works for any number of photoreceptors, as long as the long-wavelength sensitivity tails are sufficiently separate, because then selective adaptation can be achieved by choosing the proper adapting light wavelengths.
L70 ‘Akaike’s (1974) information Criterion (AIC)’ – there is no reference Akaike (1974). Maybe also refer to Symonds and Moussalli, Behav Ecol Sociobiol (2011) 65:13–21
L71 ‘relative contribution’ – contribution to…?
L72, 77, 79, 81 What is it, absorbance or absorptance, or both?
L79 I don’t know what ‘a cross-sectional photoreceptor array’ is.
L86 Here lj seems to be misplaced, as in Eq. 2 it is defined as the length of photoreceptor j.
L88 Ai/A i index
L95 The fraction of light transmitted is called the transmittance (as in L115), commonly given by parameter T. Use that, by saying for instance T0 = 1.
L99 ‘global model then only alpha bands were considered’ – sentence?
L104 ‘dark-acclimated organisms’ – better ‘dark-adapted eyes’. Idem L213, 215, 241.
L109, 117, 119 ‘possess’ vs ‘possesses’?
L124 and throughout paper: adjust formatting of references: from Arikawa et al. (1087), Beckmann et al. (2015),….
L127, 128 and throughout paper: add space between value and dimension > 10 nm, 690 nm
L141 Eq. 2,
L142 That sentence needs scrutiny concerning plural/singular: forms, a longer
L151 Check formula: parenthesis, comma
L153 λmax not defined; Ai/A print italics
L160 K italics
L163 Write, for readability, Eq. 5 as numerator/denominator
L173 ‘Model results indicated beta bands were uninformative’ will be fully uninformative for most readers; idem with ‘least important covariate β’
L180 Here the author plunges into the results without any proper description of the spectra, some repetition of the modeling applied, some intuitive explanations why this works.
L189, 194 The models are absorption not absorptance models; absorption concerns the phenomenon, absorptance is a parameter defined in physical optics.
L252 I don’t have a feeling how ‘future opsin expression studies’ could benefit.
The references need some care: remove superfluous capitals (In Arendt, Arnold, Cronin, Kelber, Posada, Wakakuwa,…); species italics Papilio (Arikawa), Drosophila (Kelber), Daphnia (Smith – delete Neuroethol. Sensory, Neural, Behav. Physiol.), …

Legend Fig. 1F Don’t understand the ‘Two-dimensional absorbance (given by eq.1)’

·

Basic reporting

In one sense, this is a well written manuscript, the English is good and in sub-sections the narrative well structured. The basic structure is fine and Tables and Figs well presented.

There are a few too many "I did"s etc and while some of these are fine, change a few to "it was found that" etc - this is a delicate balance of abstract and boring v personal and engaging.

The main problem in terms of reporting and clarity is that it is not clear what is being attempted here, until near the end of the manuscript. This comes through in several sentences from the Abstract on. What is the main aim here? It is the introduction of a new and hoped-for more accurate method of dealing with extracellularly measured overall spectral sensitivity data from eg ERGs?

Within the introduction there are useful comments about how to measure spectral sensitivity but, I at least became immediately confused about what the AIC was being aimed at? Extracellular and Intracellular methods are mentioned but while we are told what gets incorporated in the model, we are not explicitely shown what data it is testing relative to these methods until later on.

A major omission here also - in the general discussion of how to measure spectral sensitivity is visual pigment extract and more importantly microspectrophotometry (MSP) where absorbances are measured. There are many papers in the literature on how this data is then combined with photoreceptor anatomy and filters in optics to construct individual and whole-eye spectral sensitivities and is the main reason that visual pigment templates were worked out in the first place. To place this work in context, this is a large omission.

So - in summary - I think there is something worth publishing here as the method does work well in the end - an explicit statement of what is being attempted and some background to measuring spectral sensitivity is missing.

Experimental design

No experiments as such were undertaken so see comments below.
In a future version I would like to see a test of this method v others and in a way - for a theoretical methods advance paper - which this is aiming to be - that is an experiment.

Validity of the findings

As above - the finding s seem to be valid and this method does toke into account multiple factors.

Aside from the above manuscript-structure and background critique, what is also missing is a good comparison of previous estimates and methods? This is I think attempted in Fig 1 but again - like the main text - what Fig. 1 is showing is a bit cryptic. A better description of what each lie is is needed.

To be honest - previous methods have and do take into account all of the factors of the AIC method - just a bit less organised. So - I need more convincing - by using a more explicit comparison and by having more clarity to start with - how this method of analysis is a real advance on previous methods.

As the author identifies, there is now a whole load of opsin data emerging and hypotheses re spectral sensitivities and colour vision in animals have hit an all-time-low in terms of validity of hypothesis and sensibility. So - this well formulated and parsimonious method COULD be a great contribution to see through the ependorf darkly.

I would love to see a re-go, however only after a rather major revision.

Additional comments

As above

---

## Round 0.2 · Minor Revisions

We are nearly there. The reviewer is generally content with your manuscript, except for the disagreement over the use of the terms absorbance and absorption. Please look at this again. The reviewer is one of the most long established and respected visual scientists in the world. Also please attend to his other general comments re references. In addition I have made a few comments on a PDF.

Reviewer 1 ·

Basic reporting

The paper has been improved, but there are number of remarks.

Experimental design

OK

Validity of the findings

OK

Additional comments

The author criticises my point about the used terms. ‘There is, however, an important term for which we do not agree: absorption vs. absorptance. Absorption and absorptance are synonymous mathematical terms, and are central to the modeling in this paper. I disagree with the reviewer’s statements that models based on Eq. [2] in the present paper, are “absorption not absorptance models”, and the reviewer’s usage of the term absorbance.’
Johnsen indeed ‘defines absorptance, and absorbance explicitly’, but the author only elaborates on absorptance. My point concerns, however, the usage of absorbance and absorption. As Johnsen explains, absorbance is –log10 of the fraction transmited light. The term absorbance should only used in that way. Absorption is a general term describing the phenomenon of ight being absorbed. We therefore say that we consider absorption models, or, we model the absorptance. The statement ‘model the absorbance of visual pigments’ is also not acceptable, see the definition of absorbance. This is all not crucially important, but a clean language improves understanding. Sentences like ‘The goal was to map relative opsin expression of each opsin to visual pigment concentration.’ are for me not the epitome of clarity. And what concerns: ‘It identified…’
Line 94: What is ‘different photoreceptor classes in cross-section to spectral absorbance’ …which cross-section? Spectral absorbance???
Line 101: For the last time: Absorption, not absorbance
Line 111: Difficult parentheses (also elsewhere). Perhaps: ‘Normalized absorbance templates developed by Stavenga, Smits & Hoenders (1993), referred to here as SSH, and by Govardovskii et al. (2000), referred to here as GFKRD,’
Line 131: possesses?
Line 296: ‘…must be employed judiciously in based on what is known for a given visual system.’ ???

Check references. The Cronin et al. book appears twice; species in italics: Fundulus heteroclitus (and many more); also references with capitals and without; journal names in full and abbreviated; PRSB in various forms; Martin ref. incomplete, idem Molstadt, idem Porter.

---

## Round 0.3 · accepted · Accept

Thank you for the way in which you have dealt with the reviewers comments. I'm very pleased to see this paper Accepted.